# Introducing Artificial Intelligence in Interpretation of Foetal Cardiotocography: Medical Dataset Curation and Preliminary Coding—An Interdisciplinary Project

**DOI:** 10.3390/mps7010005

**Published:** 2024-01-04

**Authors:** Jasmin Leonie Aeberhard, Anda-Petronela Radan, Ramin Abolfazl Soltani, Karin Maya Strahm, Sophie Schneider, Adriana Carrié, Mathieu Lemay, Jens Krauss, Ricard Delgado-Gonzalo, Daniel Surbek

**Affiliations:** 1Medical Faculty, University of Bern, 3010 Bern, Switzerland; jasmin-aeberhard@bluewin.ch; 2Department of Obstetrics and Gynecology, Bern University Hospital, Insel Hospital, University of Bern, Friedbühlstrasse 19, 3010 Bern, Switzerland; 3Centre Suisse d’Électronique et de Microtechnique CSEM, Rue Jaquet-Droz 1, 2002 Neuchâtel, Switzerland

**Keywords:** obstetrics, cardiotocography (CTG), foetal monitoring, artificial intelligence (AI), machine learning (ML), deep learning (DL), neural network (NN)

## Abstract

Artificial intelligence (AI) is gaining increasing interest in the field of medicine because of its capacity to process big data and pattern recognition. Cardiotocography (CTG) is widely used for the assessment of foetal well-being and uterine contractions during pregnancy and labour. It is characterised by inter- and intraobserver variability in interpretation, which depends on the observers’ experience. Artificial intelligence (AI)-assisted interpretation could improve its quality and, thus, intrapartal care. Cardiotocography (CTG) raw signals from labouring women were extracted from the database at the University Hospital of Bern between 2006 and 2019. Later, they were matched with the corresponding foetal outcomes, namely arterial umbilical cord pH and 5-min APGAR score. Excluded were deliveries where data were incomplete, as well as multiple births. Clinical data were grouped regarding foetal pH and APGAR score at 5 min after delivery. Physiological foetal pH was defined as 7.15 and above, and a 5-min APGAR score was considered physiologic when reaching ≥7. With these groups, the algorithm was trained to predict foetal hypoxia. Raw data from 19,399 CTG recordings could be exported. This was accomplished by manually searching the patient’s identification numbers (PIDs) and extracting the corresponding raw data from each episode. For some patients, only one episode per pregnancy could be found, whereas for others, up to ten episodes were available. Initially, 3400 corresponding clinical outcomes were found for the 19,399 CTGs (17.52%). Due to the small size, this dataset was rejected, and a new search strategy was elaborated. After further matching and curation, 6141 (31.65%) paired data samples could be extracted (cardiotocography raw data and corresponding maternal and foetal outcomes). Of these, half will be used to train artificial intelligence (AI) algorithms, whereas the other half will be used for analysis of efficacy. Complete data could only be found for one-third of the available population. Yet, to our knowledge, this is the most exhaustive and second-largest cardiotocography database worldwide, which can be used for computer analysis and programming. A further enrichment of the database is planned.

## 1. Introduction

Artificial Intelligence (AI) encompasses a wide variety of technologies with cognitive functions like problem-solving and learning. AI has become a part of our everyday life in the form of drones, smartphones, and virtual assistants. Moreover, it is gaining interest in the field of medicine in the form of physical objects like “carebots” [1] or surgery assistants [2], as well as in virtual form represented by generative artificial intelligence [3] and machine learning (ML) [4]. ML and deep learning (DL) [5] systems learn through experience. These systems are able to recognise patterns, classify sequences based on algorithms, and form a strategy to solve problems [6]. Processing and learning from big data are other main factors in AI systems [7].

The main foetal surveillance method in obstetrics is cardiotocography (CTG). It records the foetal heart rate (FHR) and uterine contractions (UC), as well as their temporal relation. Furthermore, it represents an assessment of the foetal state before and during labour. Physicians decide on further management based on the CTG signals to prevent poor neonatal outcomes like neonatal acidosis or hypoxia, stillbirth, and cerebral palsy [8]. The impact of CTG is widely discussed because of intra- and interobserver variability in interpretation as well as low specificity. Since its introduction in the 1960s, caesarean section rates and instrumental vaginal births have increased, but no clear reduction in poor neonatal outcomes has been achieved [9]. The observer’s experience has a great influence on the quality of CTG interpretation, which offers a suitable background to implement AI-assisted interpretation. Especially human factors like long hours of work, less time to interpret because of a big workload, or less experience play an important role in the quality of interpretation. AI is not impacted by these factors [10].

In the field of radiology, studies have shown that AI systems interpret images in the same variability range as experienced specialists [11,12]. Another study shows the high performance of a decision support system in foetal ultrasound examination [13]. Since CTG interpretation is also image-based, it is hoped that AI-assisted interpretation leads to an improvement in the quality of the interpretation. With an improvement in CTG interpretation, a lower rate of unnecessary medical interventions and ameliorated intrapartal care is expected [10].

In this area, literature is still scarce. Several commercially available systems exist, with equal efficiency compared to human interpretation, yet none of these has gained general acceptance or widespread use. Solely, the Omniview–Sisporto system could be proven to be superior to expert interpretation so far, yet in one single study with retrospective design [14]. With AI algorithms developing and improving, new approaches in AI-assisted CTG interpretation are being studied. It is generally known that the bigger the data, the better the AI system functions. It is a general purpose of every AI project to generate a large dataset on which the algorithms are based. Furthermore, data has to be accurate, include no artefacts, and be homogenous.

The objective of this manuscript is to obtain dataset curation prior to programming, which was accomplished by the author within the Centre Suisse d’Electronique et de Microtechnique (CSEM). The final objective is to develop an AI system for CTG interpretation and thus improve foetal outcomes by reducing inter- and intraobserver variability in CTG interpretation. The authors intend to develop an AI system for CTG interpretation and thus improve foetal outcome by reducing inter- and intraobserver variability in interpretation. The authors do not intend to measure inter- and intraobserver variability itself but to compare foetal outcomes with and without additional use of the created AI system in a future prospective randomised clinical trial.

## 2. Material and Methods

We conducted a retrospective study as a collaboration between the obstetrics department at the University Hospital of Bern (UHB) and Centre Suisse d’Electronique et de Microtechnique in Neuchâtel, Switzerland (CSEM).

The author participated in data curation at the UHB, particularly by accomplishing data export and anonymisation. The methodology will be described as follows.

Following this, we describe the medical system Intellispace perinatal [15], which is used in our hospital for CTG monitoring, data collection, and storage.

## 3. Intellispace Perinatal

Intellispace perinatal [15] is a production of PHILIPS [16], the Department of Obstetrics, and the Laboratory of Computer Science of the Massachusetts General Hospitals. It allows documentation and monitoring of the foetus and mother before, during, and after birth. The manufacturers claim easy access to the documentation from multiple sources due to the fact that Intellispace can be easily integrated into different clinical software systems. Furthermore, it functions based on an alarm system that prevents the user when a critical event occurs. The algorithms detect baseline changes, variability, accelerations, decelerations, and contractions as defined by the National Institute of Child Health and Human Development (NICHD) [17]. Critical events include foetal tachycardia or bradycardia, signal loss, low or absent variability, and decelerations.

A medical file is created for each patient (mother) and saved in the system. One file can contain several episodes. An episode represents a recording for a specific period of time. As the system can be used ante-, intra-, and postpartum, each woman can present different episodes originating from one pregnancy, as well as different episodes from different pregnancies. This is highly dependent on the need to monitor of a pregnancy.

The top of the interface depicts the patient data, the actual date, and the time. The alarms are coded as a bell in different colours. In the middle, the foetal heart rate (FHR) is shown with a numeric scale in beats per minute (bpm). The maternal heart rate (MHR) is illustrated in a different colour. If there are multiple foetuses, each FHR is represented with a different colour. Below the heart rate, foetal movements are pictured as black bars. Maternal contractions appear with their numeric scale in millimetres of mercury (mmHg). As the heart rates and the contractions are shown on the same timeline, relationships between the FHR and the maternal contractions can be seen directly. Figure 1 depicts a graphical representation of a CTG, as it is seen in maternity wards on a daily basis.

## 4. Data Collecting

Data collection was conducted on two pillars: CTG data and corresponding clinical data.

## 5. Defining the Cohort and Collection of Clinical Data

For obtaining a complete overview of the deliveries during the predefined study period, different sources were considered: the written archives of the delivery ward (the so-called “delivery books”), the digital database of the puerperium department, and/or the digital database of the patient management department. Finally, we opted for the patient management database, which proved to be the most complete. The written archive could have potentially represented an even wider source of data. However, manually searching for every patient in a written book would have been too labour-intensive, so this method was excluded due to a lack of efficiency. The digital database of the puerperium department only started to be written in 2016, so this option was excluded from the start due to its limited size. The final study sample was listed in tabular form with a tab for each year.

The collection of outcomes corresponding to each mother–foetus pair started in January 2021 with an application for foetal and maternal outcomes to the Insel Data Centre (IDC). A Data Transfer Agreement between CSEM and the UHB was created in March 2021 in collaboration with legal advisors at Unitectra. In June 2021, the IDC reported missing elements in the patient identification numbers (PIDs), yet correct identification of the patients was still possible. Correct identity was confirmed by the study team. The purpose of obtaining clinical data was to analyse foetal outcomes corresponding to each CTG, which is indispensable for training the AI system. The main foetal outcomes of interest were umbilical cord arterial pH as well as a five-minute APGAR score. These outcomes helped assess foetal asphyxia during labour and the clinical adaptation of the newborn. Clinical data were collected from the digital archives of the University Hospital of Bern (UHB), Switzerland, and included all deliveries between 2006 and 2019.

## 6. CTG Data

CTG data assessment was conducted in IntelliSpace Perinatal [15]. Although exclusively graphical data are being used in the daily routine, this is not sufficiently accurate for programming an AI system. Thus, raw data had to be extracted for each CTG. To achieve data extraction, an update of the Intellispace perinatal [15] software was necessary in the first phase (version Rev. K). For the initial extraction process, the author searched patients by their names, surnames, and date of birth in the Intellispace system [15]. This process led to a large amount of missing data because of spelling discrepancies/mistakes. The search was restarted by means of PID, which consistently increased the accuracy of the search.

For the purpose of downloading raw data, a shared folder was created by the informatics department at our hospital, with the support of the Philips [16] technicians. Access was exclusive to study personnel. 

## 7. Eligibility Criteria

We included CTG files from the obstetrics department of the University Hospital of Bern (UHB) in our database, as well as the corresponding foetal outcomes. The CTG files were from women who delivered from 2006 to 2019 at our institution. Only peripartal CTG files have been included.

Excluded were births from patients who did not give their consent to use their data, as well as patients for whom the data were incomplete. Multiple births were excluded. 

Physiological foetal pH was defined as 7.15 and above; the five-minute APGAR score was considered physiologic when reaching ≥7. There is no generally accepted definition of hypoxia internationally, so the authors decided on the values corresponding to the internal hospital guidelines.

## 8. Results

### 8.1. CTG Data

Raw data from 19,399 CTG recordings could be exported. This was accomplished in the CTG recording system Intellispace Perinatal [15] by manually searching PIDs and downloading the corresponding raw data of each episode. For some patients, only one episode per pregnancy could be found, whereas for others, up to ten episodes were available. We did not statistically analyse the number of episodes per patient since this was not relevant to our study. For each patient, the episode corresponding to the latest time point in the current pregnancy was extracted because this always corresponded to the labour period.

At the beginning of the Methods section, we have depicted a graphical image of the CTG, which is commonly used by midwives and obstetricians for interpretation. As follows, we present an example of raw data depiction (“the data behind the data”). These appear in an Excel document containing numbers, which represent data points of the CTG. An example is shown in Figure 2 to illustrate this. It is important to mention that this is an incomplete data example because of the enormous size of these documents.

In Table 1, a listing of manually extracted CTG episodes per year is shown.

### 8.2. Clinical Data

For the 19,399 CTG documents, clinical outcomes were searched using the IDC. In August 2021, IDC provided the study team with the first data sample, which contained 3400 outcomes. Due to the small size of the cohort where clinical outcomes were available, the dataset was rejected by the study team, and a new search strategy was elaborated together with the IDC. The authors used control measures for the quality of the dataset, including the number of cases provided, completeness of data (meaning a set of CTG and complete corresponding clinical parameters), availability of signed general consent where applicable, and the presence of both physiological and pathological clinical outcomes following a standard distribution.

The initial small sample size resulted from a high number of search criteria provided to the IDC. The most important limitation factor was the delivery stage, which was defined by the clinician team using two landmarks: the opening of the cervix at four and ten centimetres. This led to an extremely time-consuming search and a limited number of samples. Moreover, information about the use of peridural anaesthesia (PDA) was inconsistent and was finally not considered for data curation. Although it represented a further limitation search factor, the delivery mode was finally coded in abbreviations and kept for the search. The IDC proposed to provide a separate list with the diagnosis at discharge, which was accepted.

Finally, using the second strategy, a cohort of 15,744 patients with available corresponding clinical outcomes could be obtained. An extensive list of assessed neonatal outcomes is depicted in Table A1 in the Appendix A.

Eventually, the data were anonymised for further use by creating pseudo-PIDs. After the first attempt of manual anonymisation was proven too extensive, a computer algorithm was created to facilitate anonymisation and data matching (CTG raw data with clinical outcomes). Finally, after exclusion from duplicates, multiple births, and datasets with missing information, as well as matching the foetal outcomes with the corresponding CTG file, a dataset of 6141 complete data samples was obtained, as shown in Figure 3.

## 9. Discussion

To our knowledge, we curated the most exhaustive as well as the second-largest CTG and foetal outcome database worldwide. Its purpose is computer analysis and programming for AI-based interpretation. Furthermore, the database distinguishes itself from all the internationally existing ones by its complexity and completeness [18].

The currently largest existing database was created within the controverted INFANT study conducted at Oxford University [19]. This includes outcomes from 47,062 women (and the corresponding 47,648 infants). The exact content of this database cannot be ascertained since it is not publicly available. It was used to obtain an AI system that finally could not achieve better neonatal outcomes than observer interpretation [19]. The INFANT study was controversial because of its design weakness: Robert Keith and others raised the concern that cross-over effects could have been present because of unblinded co-located clinicians [20].

Our database is currently not publicly available. To our knowledge, two public databases exist: the CTU-UHB database from the Czech Republic [21] and the UCI repository [22].

The CTU-UHB database contains 552 intrapartal CTG tracings. The tracings were assessed at the University Hospital of Brno between 2010 and 2012. Although it contains complete CTG episodes, the CTG sets in this database are characterized by an important number of missing data points, especially at the end of labour. This is common if the cardiotocograph cannot receive a signal because the detector is being moved, for example, by movements or other disturbance factors. Missing data points make the AI-assisted analysis of the CTGs more difficult, which could have an influence on performance results [21]. Compared to our database, it comprises considerably fewer data samples. As a reminder, AI systems work best when trained with a large amount of data.

The UCI machine learning repository contains 2126 CTG tracings from the University of California Irvine. Three expert obstetricians classified the CTG tracings in addition to automatic processing [22]. This database was used by Ayres de Campos et al. to create SisPorto 2.0 [23]. As mentioned before, SisPorto is currently the most promising AI system for CTG interpretation, yet it has not reached extensive market penetration so far.

AI-assisted interpretation of CTG could support clinical reasoning during delivery. There is an ongoing debate concerning the possibilities of improving neonatal outcomes. Improve CTG interpretation is one way to advance in intrapartum care, as foetal hypoxia prediction during birth could support clinical decision-making. Reversely, the use of CTG itself has been widely discussed and debated ever since its introduction decades ago. 

The size of the created database is a strength of our study. Yet, the potential of the assessment at this study start was substantially higher, as only a third of the initial candidates could be selected for final introduction into the dataset. This is due to data loss, duplication, incorrect assessment, etc. This underlines the difficulty of data curation and creation of large datasets with patients’ outcomes originating from different sources, which often leads to underpowered analyses. 

One limitation of this study could be the risk of bias due to low recruitment in certain population categories, such as patients with a language barrier where general consent is not collected because explanations cannot be properly provided. Other biases are represented by limited recruitment in the first years of this study, where the electronic database had just been implemented, and data loss was still very high. However, these biases can be balanced, in our opinion, by the large number of patients included.

The main challenge of the dataset curation in our study was obtaining the raw data for CTGs. This required extensive and time-consuming preparation, an important update of the entire Intellispace system [15], and was associated with high costs. Taking a glance at the size of one single episode of raw data, it is easy to understand why the observer’s eye is not able to interpret every single variation in foetal heart rate, especially in correlation with further features such as the presence of contractions and the delivery stage.

The “data behind the data” and its computerised interpretation show immense potential for computer-assisted AI interpretation, yet considering the current advances in the medical field in regard to artificial intelligence, foetal surveillance is still making “baby steps” in taking advantage of this trend.

While the use of continuous foetal surveillance during labour is an issue being debated in its own right [9], the authors of this study believe that withholding information from the observer is not the solution but rather a refinement of interpretation algorithms, whether based on AI or other models [24].

Our project is still ongoing. An extension of the database was approved by the Ethical Committee of Bern and is currently under work (deliveries up to April 2022). After further refinement, the dataset will be used to program and train an AI algorithm for CTG interpretation, with the intention to test its efficacy in a future randomised controlled trial (RCT).

## 10. Conclusions

Our work reveals how extensive data curation can be and brings a new perspective for clinicians mainly working on projects of a clinical or laboratory nature. This endeavour underlines the importance of interdisciplinary collaborations for the medical practice of the future, where boundaries between biology and technology, thus human and computer, are getting fainter as we advance through the century, hopefully only to our best. 

Even though bigger data are always better for AI interpretation, our curation is, to our knowledge, the second-largest CTG database worldwide and distinguishes itself by completeness and complexity, being the most exhaustive of its type currently available. 

## Figures and Tables

**Figure 1 mps-07-00005-f001:**
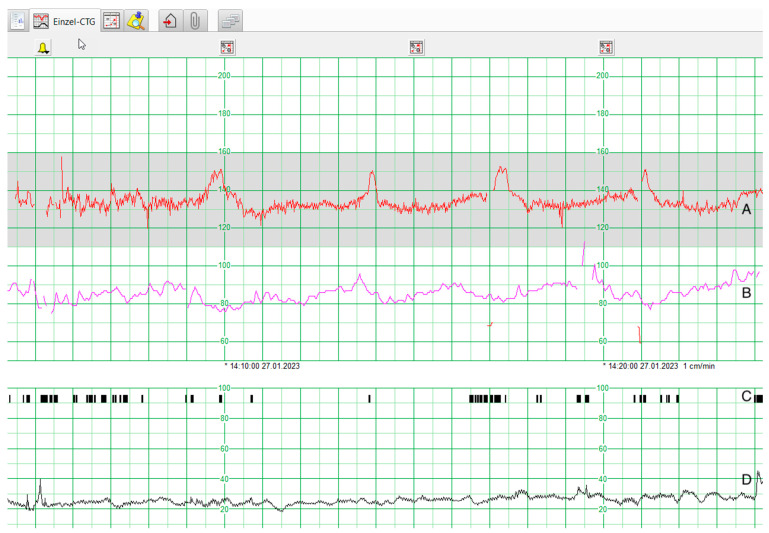
Graphical representation of a cardiotocogram in Intellispace Perinatal [15]. (**A**) Foetal heart rate (FHR). (**B**) Maternal heart rate (MHR). (**C**) Foetal movement. (**D**) Uterine contractions.

**Figure 2 mps-07-00005-f002:**
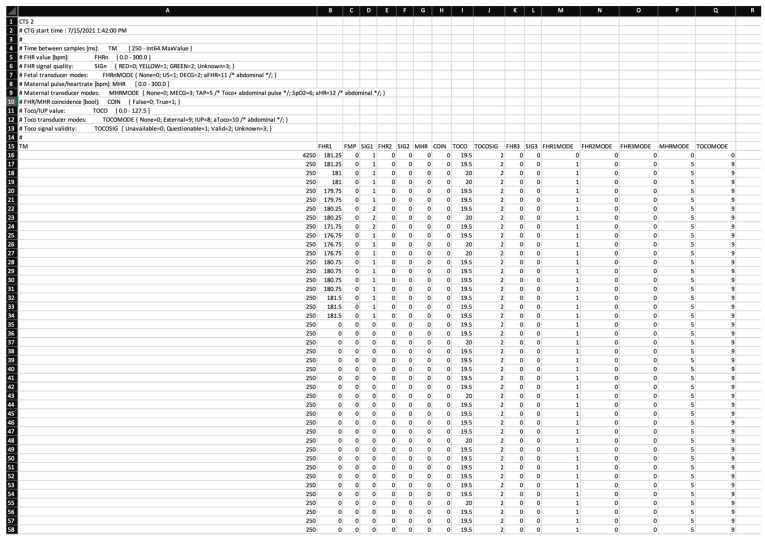
Example of raw data from a cardiotocography tracing.

**Figure 3 mps-07-00005-f003:**
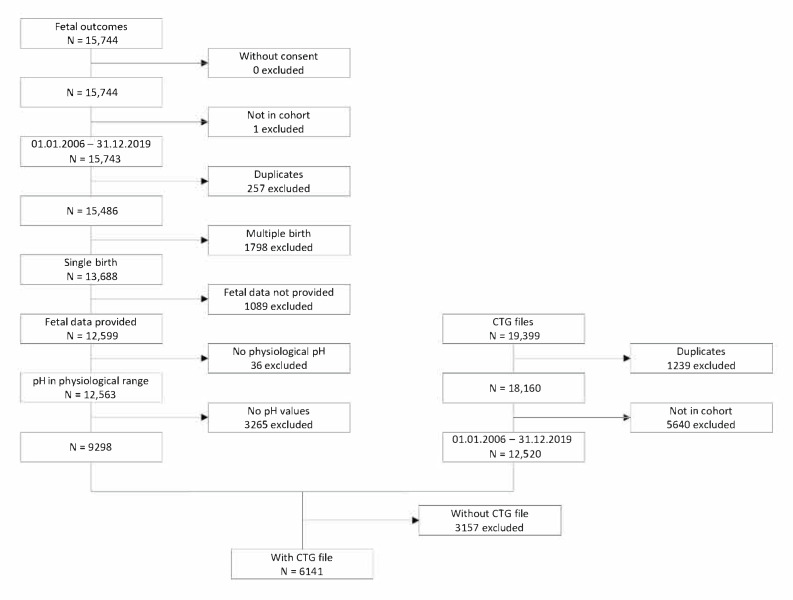
Selection of collected data (Flowchart, Centre Suisse d’Electronique et de Microtechnique CSEM SA, 2021).

**Table 1 mps-07-00005-t001:** Women per year manually extracted from the Intellispace perinatal system at the University Hospital of Bern.

Year	Number of CTG Episodes
2006	0
2007	204
2008	376
2009	526
2010	542
2011	1690
2012	1742
2013	1820
2014	1800
2015	2294
2016	2888
2017	345
2018	271
2019	164

## Data Availability

The datasets generated and analysed during the current study are not yet publicly available due to confidentiality agreements, as well as ongoing curation of the data. Additional information will be available from the corresponding author upon reasonable request.

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
