# Peer review of "Introducing Artificial Intelligence in Interpretation of Foetal Cardiotocography: Medical Dataset Curation and Preliminary Coding—An Interdisciplinary Project"

_mps, 2024, doi:10.3390/mps7010005_

Round 1

Reviewer 1 Report

Comments and Suggestions for Authors

My overall opinion is that this protocol and these preliminary data are comprehensive and very well presented. The introduction focuses well on the problem of inter-observer variability in the interpretation of CTGs, the need to improve this variability, and the possibilities of AI for achieving this. The methods are described in detail, providing an understanding of how recruitment, data extraction and outcomes will be carried out. The two outcome research strategies, and the limitations of the first, are presented transparently. The discussion takes up the strengths and weaknesses of this database, and reinforces the importance of using AI to improve the interpretation of CTGs.

Here are a few minor comments that might be worth addressing before publication of this manuscript if they seem relevant to the authors.

Abstract :

As Methods and Protocols is an international journal with a wide readership, and not specialised in obstetrics, I would add in the background that CTG is widely used during pregnancy as a method of assessing foetal well-being and uterine contractions.

In the methods, I would specify whether you included only CTGs during labour or also CTGs for monitoring high-risk pregnancies before labour.

In the discussion, it might be interesting to explain what biases are expected with regard to unavailable outcomes, and what strategy you plan to mitigate these biases. For example, if the unavailable outcomes are due to patients who have been re-transferred to low-risk hospitals after follow-up in a university hospital, their outcomes will probably be more favourable than a cohort of patients who remain monitored in an academic centre. This means that you could unintentionally select CTGs from pregnancies with a high risk of complications and fetal hypo/asphyxia.

Introduction

In the opening paragraph, it might be interesting to add generative AI to the list of potential uses of AI (for counselling during pregnancy for example).

In the paragraph on the use of AI in radiology (l.50), you could also mention the performance of AI for obstetrical ultrasound (ref: https://doi.org/10.1002/uog.26242).

Methods

As in the abstract, it would be useful for readers to specify whether you have included only per-partum or also ante-partum CTGs.

Results

Table 1: How do you explain the fact that you have ten times more episodes between 2011 and 2016 than in other years?

Discussion

As in the abstract, it would be interesting to present the expected biases and the strategies you plan to use to mitigate them.

Reviewer 2 Report

Comments and Suggestions for Authors

At the end of the "Introduction" chapter authors write:

The objective of this manuscript is to obtain dataset curation prior to programming, which was accomplished by the author within the Centre Suisse d’Electronique et de Microtechnique (CSEM). The final objective is to develop an AI-system for CTG interpretation and thus improve foetal outcome, by reducing inter- and intraobserver variability in CTG interpretation.

The first goal was achieved. What about the final objective? Of course, it is not possible to give a complete answer to this question at the present time. However authors should address in which way they intend to measure inter- and intraobserver variability.

The final objective is to develop an AI-system for CTG interpreta-66
tion and thus improve foetal outcome, by reducing inter- and intraobserver variability in67
CTG interpretation.

Reviewer 3 Report

Comments and Suggestions for Authors

1.       The abstract provides a clear overview of the study's background, methods, results, and discussion. However, the information could be more logically organized for enhanced readability.  Consider rephrasing and breaking down complex sentences to improve clarity.

2.       The introduction section effectively outlines the significance of AI in medicine, particularly in the context of cardiotocography. However, it would benefit from a more explicit statement on the existing challenges in manual interpretation that AI aims to address.

3.       The methods section adequately describes the dataset extraction process. However, details on the rationale for excluding certain deliveries and the criteria used for grouping clinical data could be more explicit. Clarify the reasons behind choosing specific cut-off values for physiological foetal pH and APGAR score.

4.        The rejection of a portion of the dataset due to small size is mentioned, but the criteria for this decision need more clarity. Provide details on the quality control measures implemented during the dataset curation process.

5.       This work presents the number of CTG recordings and corresponding outcomes, but lacks details on the characteristics of the population studied. Elaborate on any unexpected findings or challenges encountered during the data extraction and matching process.

6.       Any potential biases during data curation and potential limitations in the dataset.

7.       The overall language is clear, but some sentences are quite dense. Consider simplifying complex sentences for a wider audience.

Comments on the Quality of English Language

 Minor editing of English language required
